# Content of Phenolic Compounds and Antioxidant Activity of New Gluten-Free Pasta with the Addition of Chestnut Flour

**DOI:** 10.3390/molecules24142623

**Published:** 2019-07-18

**Authors:** Anna Oniszczuk, Gabriela Widelska, Agnieszka Wójtowicz, Tomasz Oniszczuk, Karolina Wojtunik-Kulesza, Ahlem Dib, Arkadiusz Matwijczuk

**Affiliations:** 1Department of Inorganic Chemistry, Medical University of Lublin, Chodźki 4a, 20-093 Lublin, Poland; 2Department of Thermal Technology and Food Process Engineering, University of Life Sciences in Lublin, Głęboka 31, 20-612 Lublin, Poland; 3Laboratoire de Nutrition et Technologie Alimentaire, Institut de la Nutrition, de l’Alimentation et des Technologies Agro-Alimentaires, Université des Frères Mentouri, 25000 Constantine 1, Algeria; 4Department of Biophysics, University of Life Sciences in Lublin, Akademicka 13, 20-950 Lublin, Poland

**Keywords:** liquid chromatography, functional food, gluten-free pasta, chestnut flour, antioxidant activity, phenolic acids

## Abstract

Chestnut fruit abounds in carbohydrates, proteins, unsaturated fatty acids, fiber, polyphenolic compounds, as well as vitamins and micronutrients, that are behind the health-promoting properties of this plant. The purpose of the discussed research was to obtain innovative gluten-free pasta from rice and field bean flour enriched with a various addition of chestnut flour. Regarding the studied pasta, the following were determined: the content of free phenolic acids, total polyphenols, and antioxidant properties. Chromatographic analysis (HPLC-ESI-MS/MS (high-performance liquid chromatography-electrospray ionization tandem mass spectrometry)) revealed a wide variety of phenolic acids. In a sample with 20% and higher content of chestnut flour, as many as 13 acids were detected. Isoferulic acid prevailed. The total content of free phenolic acids and total polyphenols increased along with the increasing chestnut content. Moreover, in most cases, the content of individual acids increased with the addition of chestnut flour. Besides, the antioxidant activity was positively correlated with the addition of chestnut fruit flour, the content of free phenolic acids, and total polyphenols. Our research has demonstrated that our innovative gluten-free pasta, with the addition of chestnut flour, has a potential to be a source of polyphenolic compounds, including free phenolic acids, that are valuable for human health.

## 1. Introduction

Around 1% of the world population is affected by celiac disease; at the same time, gluten-free or gluten-low diets are not in fashion [1]. Celiac people are forced to follow a strict gluten-free diet, which is often unbalanced and lacks many nutrients [2,3]. Gluten-sensitive consumers are looking for valuable food products because gluten-free foods are likely to have a decreased nutritional quality as compared to their gluten-rich equivalents [4]. In recent years, the market for gluten-free products has been growing exponentially and constantly, but there is still a need to provide new and better quality products to celiac people. The market value of this sector, in terms of volume, was estimated to be 393.43 kt in 2015 and was projected to grow at the CAGR (compound annual growth rate) of 10.4% in 2020 [5]. The manufacturing of gluten-free products is challenging as gluten determines some of the most essential structural, rheological, and organoleptic properties of bread, snacks, or pasta. Currently, no direct substitute for gluten is available. Instead, a combination of refined unfortified cereal flours (e.g., maize and rice), some hydrocolloids, and proteins are used to produce gluten-free foods. Additionally, the cost of gluten-free pasta may be up to 70% higher compared to standard pasta [6]. Acceptable gluten-free products may be designed using alternative flours that may also enhance the nutritional quality and the glycemic index of such foodstuff. Allen and Orfila [6] found that the protein content in gluten-free pasta was six times lower than in conventional pasta. For example, the addition of blue maize flour (rich in polyphenols) to gluten-free pasta at 50 and 75% increased the levels of protein, dietary fiber, and bioactive compounds with antioxidant capacity, thereby increasing the slowly digestible and resistant starch fractions [7]. The application of legume flours in the amount of up to 30% increased the phenolics composition in gluten-free rice-based pasta with no negative effect on the cooking characteristics [8]. However, gluten-free cereals, legumes, and pseudocereals may differ significantly across samples containing phenolic compounds and an antioxidant profile [1,4,7]. Studies on the nutritional content of gluten-free products and the quality of celiac people’s diets are scarce, and any presented results seem to highlight the need for a more careful product formulation and dietary guidance for celiac consumers [9]. 

Consumers are increasingly discerning regarding food safety, composition, and health-related issues [10]. One of the fastest-growing markets is functional or novel foods, and consumers are likely to buy products with specific functional properties [11]. Consumer’s acceptance of functional food is associated with processing technologies, scientific innovations, and their health status, especially when food may deliver real health benefits to people at risk of, or suffering from, major degenerative, chronic, or gastrointestinal diseases [11,12]. Modification of staple foods has been perceived as beneficial, and consumers tend to prefer processes, such as fortification and traditional cross-breeding, to increase the nutritional value of, especially, healthy cereal products [13]. The growing awareness of the link between nutrition and health encourages the pursuit of health-promoting supplements abounding in bioactive compounds as a way to counteract an unbalanced diet, especially free of gluten [10,14].

Polyphenols are among the most important phytochemicals in nutritional terms, as they contain mainly flavonoids and phenolic acids. There are over 8000 phenolic structures already discovered, and over 500 are present in plant foods as dietary polyphenols. They contain complexes with at least one or more phenolic groups in the structure. They are responsible for the strong reducing properties of polyphenols. In the natural environment, they occur as simple molecules, like phenolic acids or flavonoids, or as very complicated polymerized macromolecules, for example, tannins [12,15]. Polyphenols are secondary metabolites of plants. Some acids are present in nearly all plants. Caffeic, *p*-coumaric, vanillic, ferulic, protocatechuic, and other acids are found in selected foods or plants (e.g., gentisic, syringic) [15]. They comprise several antioxidant compounds and are generally considered to be involved in the prevention or treatment of chronic diseases in humans [16]. Experimental studies on animals or cultured human cell lines confirm the role of polyphenols in the prevention of cardiovascular diseases, cancers, neurodegenerative diseases, diabetes, or osteoporosis [17]. 

Given that, the chestnut fruit (*Castanea sativa* Mill.) from the beech family *Fagaceae* appears to be a viable solution [18]. It is used in the production of marron flour and chestnut flour as a component of gluten-free diets [19]. The chestnut plant comes from Asia Minor. It was first brought to Greece and then reached France, Spain, Algeria, and Italy. The chestnut was cultivated already in ancient times [20,21,22,23]. Today, most of its species can be found in the Mediterranean countries. In Corsica and Italy, chestnuts were used for making bread and confectionery. Also today, chestnut flour is used in making bread combined with wheat or barley flour. Until the Second World War, chestnuts were common in human and animal nutrition [24,25]. Today, roasted chestnuts can be bought and eaten in the street, which is a tourist attraction in some countries. There are some reports of products based on or with the addition of chestnut flour as an additive to pasta [26], bread [27], cookies [28], snacks [29], or meat [30]. Chestnuts have become increasingly popular, especially in a gluten-free diet.

The chemical composition of chestnuts may vary depending on the cultivar, origin, country, environmental factors, growing conditions, and harvest period. As reported previously [31,32,33], starch is the predominant component in the chestnut fruit, ranging from 39 to 87 g/100 g d.w (dry weight), and its content is important in the food industry, flour production, or in animal feed. The protein content in chestnuts is similar to that in cereals (4.9–7.4 g/100 g d.m. as crude protein); the fat content (1.7–3.1 g/100 g d.w.) is much lower than in other nuts (hazelnuts or almonds), thus, making chestnuts valuable as less caloric food [31]. Chestnuts also abound in micronutrients, especially K, P, Mg, and low in Ca (45 mg/100 g d.m. on average). Concerning micronutrients, the most abundant is Fe, followed by Mn, Zn, and Cu. Also, B-, C- and K-group vitamins, as well as polyphenolic compounds, have been reported in chestnuts [26,27,28]. 

Due to the health-promoting and taste properties of chestnuts, the researchers aimed to obtain gluten-free pasta with a various level of addition of chestnut flour. Once obtained, the pasta was tested for the content of bioactive compounds, in particular, free phenolic acids and total polyphenols. The final step was to determine the antioxidant properties of the tested samples.

## 2. Results and Discussion

### 2.1. Quantitative Analysis of Phenolic Acids 

Polyphenols are the major group of biologically active non-nutrients present in food. They display physiological effects that help prevent several lifestyles and chronic diseases [12]. The qualitative and quantitative analyses of the content of phenolic acids in gluten-free pasta made from blends of rice flour with field beans enriched with the 10, 20, 30, 40, and 50% addition of chestnut flour were carried out using high-performance liquid chromatography-electrospray ionization tandem mass spectrometry (HPLC-ESI-MS/MS). The chromatographic method was validated. The content of free phenolic acids in the examined extracts was determined based on calibration curves designed for each model. All the tested compounds showed the positive linearity of matching the curve. The correlation coefficients for all calibration curves were *r^2^* > 0.9991. Calibration curve equations, the limit of detection (LOD), and the limit of quantification (LOQ), as well as the ranges of linearity for assays of individual phenolic acids, were demonstrated by the authors in a previous publication [34]. The extraction method used was 40 min ultrasonic-assisted extraction at elevated temperature (60 °C) with the use of an 80% aqueous ethanol solution. This method was selected because, in previous experiments, it had proven to be the optimum technique for the isolation of phenolic acids from food enriched with phenolics functional additives [35,36,37].

Mass spectrometry is a method used to measure molecular weight. It helps collect on the composition and, specifically, the structure of molecular compounds. The proportionality of signal intensity and the amount of analyte allow the use of mass spectrometry in quantitative assays. The development of detectors and ionization techniques and their coupling with chromatography have broadened the range of combinations, including ionic compounds or macromolecules greatly [38]. The chromatographic analysis revealed a large variety of phenolic acids in the studied gluten-free pasta. In a sample with 20% and a higher content of chestnut flour, as many as 13 acids were identified (Figure 1, Table 1). These were mostly benzoic acid derivatives: gallic, protocatechuic, 4-OH-benzoic, salicylic, vanilic, gentisic, *trans*-synapic, *cis*-synapic, and cinnamic acid derivatives: *trans*-caffeic, *cis*-caffeic, *p*-coumaric, ferulic, and isoferulic. In the pasta with the addition of 10% of chestnut, vanillic acid was not detected, while all the other above-mentioned acids were. In all the analyzed extracts of pasta enriched with chestnut flour, the caffeic acid content was above the limit of detection but below the limit of quantification. Only seven free phenolic acids (protocatechuic, 4-OH-benzoic, salicylic, gentisic, *p*-coumaric, ferulic, and isoferulic) were determined in the samples of rice-field bean pasta without the functional additive. Isoferulic acid prevailed in all the tested samples. 

Phenolic acids may differ in quality and quantity compared to those covered in our study, but some differences are found to be due to the raw materials used. This was the case for black, blue, and yellow barleys in terms of their composition of free and bound phenolic acids [39]. The same was true for pasta composition, as in the case of wheat-based barley enriched pasta [40] or rice-based yellow pea supplemented precooked pasta [8]. The total content of free phenolic acids increased along with the increase of the chestnut content in the samples and reached 34.015, 38.927, 46.977, 51.468, 56.595, and 65.010 μg/g of dry matter for the control sample without the addition and samples with the addition of 10, 20, 30 40, and 50% of chestnut flour, respectively. Durazzo et al. [41] reported the total phenolics content in sweet chestnuts as 365.80 mg/100 g in aqueous extracts and as 896.01 mg/100 g in residues after extraction, so, besides extracts, also residues could have great potential as a supplement of food products. 

Also, the content of individual acids increased as more chestnut flour was added. The 4-OH-benzoic and salicylic acids were the only exceptions: their content gradually decreased and reached the lowest value in the sample enriched with 50% of chestnut flour. This is because the basic components of non-enriched pasta, i.e., a blend of rice and field bean flour, contain more of these two acids than the chestnut fruit. It, therefore, naturally follows that the addition of chestnut flour reduced the quantity of the two compounds. The tests showed that the obtained kinds of pasta are a great source of free phenolic acids as natural antioxidants. Similarly, Bouasla et al. [8] reported 15 phenolic compounds in rice-yellow pea pasta processed with the extrusion-cooking, namely, protocatechuic, 4-OH-benzoic, vanilic, *trans*-*p*-coumaric, *cis*-*p*-coumaric, *trans*-ferulic, *cis*-ferulic, salicylic, gentisic, *trans*-caffeic, *cis*-caffeic, syringic, 3-OH-cinnamic, *trans*-sinapic, and *cis*-sinapic acids. On the other hand, *trans*- and *cis*-ferulic acids prevailed in all samples. The inclusion of whole barley flour into pasta formulations resulted in an increase of the phenolic content due to the higher content of free and bound forms of phenolic acids of barley flour (13.9 μg/g and 654.0 μg/g, respectively) than semolina (8.4 μg/g and 194.0 μg/g, respectively) [40]. It has been widely demonstrated that the consumption of foods with a high content of phenolic acids promises anti-cancer, anti-bacterial, and anti-inflammatory effects [5,42,43], especially ferulic acid has a potential to be a health-promoting component as chemoprotectant [40]. Recently, phenolics and flavonoids are reported to be great antioxidants and have proven to be more effective than vitamin C, E, and carotenoids [44]. Chestnut flour is also an important source of other nutrients, such as unsaturated fatty acids, proteins, vitamins, minerals, fiber, or polyphenols [45,46]. 

Additionally, some important changes/reduction in the composition of phenolics may be observed after thermal processing, probably due to oxidative reactions caused by water, oxygen, and heat [47]. This is important, especially for pasta products, which are consumed immediately after cooking. Zhang et al. [48] found that bound phenolics in chestnut kernels were stable. However, prolonged steaming at high temperature improved the levels of phenolics, flavonoids, as well as the antioxidant activity up to 60.11% above the original value. During steaming, seven phenolic compounds increased their value, including ferulic acid, chlorogenic acid, gallic acid, vanillic acid, syringate, *p*-coumaric acid, and quercetin. Li et al. [49] found a relatively significant effect of the thermal treatment method on the nutritional composition of chestnuts – due to the total phenolics content—if boiled, roasted, or fried. Regarding pasta composition, De Paula et al. [40] confirmed ferulic acid to be the major phenolic acid found in free and bound phenolic extracts in durum wheat pasta with added barley flour. The pasta making process (mixing and extrusion) resulted in a reduction in free phenolic acids in all fresh and uncooked pasta formulations. The observed variation in the free phenolic acids content was mainly due to a decrease in ferulic, caffeic, and *p*-hydroxybenzoic acid (insignificantly). The drying process affected the individual phenolic compounds in the free and bound fractions differently, and vanillic, caffeic, and *p*-coumaric acids did not significantly change, while *p*-hydroxybenzoic and ferulic acids increased after drying. Moreover, the cooking of pasta did not affect total phenolic acids significantly. They rather tended to conserve free and bound phenolic compounds. Kosović et al. [26] tested the cooking quality of pasta processed with an extruder and Minipress with durum wheat replaced with 10, 15, and 20% of chestnut flour pasta. They found that the addition of chestnut flour to durum wheat pasta decreased the optimum cooking time, hardness, cohesiveness, and chewiness but increased cooking losses and pasta adhesiveness. However, it must not be ignored that the pasta product developed in our study is gluten-free food and can be consumed by people suffering from celiac disease, gluten intolerance, or allergy to this nutrient [27,46,50,51,52]. 

### 2.2. Total Polyphenol Content and the Antioxidant Properties of Gluten-Free Pasta

Polyphenol compounds are a common group of secondary metabolites, and many desirable biological effects depend on whether they are present in functional food products. It is a common property that aglycones show higher antioxidant activity than glycosidic forms or ones bound by other types of bonds [53]. The antioxidant activity of phenolic acids also depends on the number of hydroxyl groups in the molecule and can be augmented by spherical effects. Cinnamic acid derivatives are more effective antioxidants than benzoic acid derivatives [54]. For that reason, in the next stage of the research, the authors determined the total polyphenol content and the 2,2-diphenyl-1-picrylhydrazyl (DPPH) free radical scavenging potential of the tested samples. The results demonstrated that the polyphenol content also raised with the increase in the level of chestnut fruit in pasta. The highest total polyphenol content was reported in a pasta sample with the addition of 50% of chestnut flour, while the lowest one was seen in pasta without this additive. Total polyphenols showed the values of 2.123, 2.621, 2.955, 3.886, 4.816, and 5.721 mg gallic acid equivalent (GAE)/g of dry matter for the control sample and samples enriched with the addition of 10, 20, 30, 40, and 50% of chestnut flour, respectively. Rocchetti et al. [4] tested various gluten-free raw materials and found that the total phenolics content varied from 52.3 and 57.0 mg GEA/100 g in white sorghum and amaranth, respectively, 275.5 mg GEA/100 g in buckwheat and up to 500.4 mg GEA/100 g in violet rice. Camelo-Mendez et al. [7] found that the total phenolics content varied from 75.4 mg GA/g in white maize, unripe plantain, and chickpea pasta methanol extracts up to 210.4 mg GA/g in blue maize, unripe plantain, and chickpea gluten-free pasta extracts followed by Folin–Ciocalteu procedure. For gluten-free dry pasta, Rocchetti et al. [55] reported higher values of the total phenolics content in the bound phenolic fraction when compared to free fractions, with GAE ranging from 7.58 mg/100 g in sorghum pasta to 32.68 mg/100 g in quinoa pasta.

Compared to the free radical DPPH, the antioxidant properties of the tested pasta were high (the percentage of DPPH sweeping after 30 minutes: 93.02, 94.11, 94.32, 94.66, 94.71, 95.16 for samples containing 0, 10, 20, 30, 40, and 50% of chestnut flour, respectively) and increased further (though slightly) as more functional additive was added (Table 2). 

Arufe et al. [28] tested the total polyphenols content and the radical scavenging activity of extracts from chestnut flour-seaweed powder blends and seaweed-enriched chestnut cookies baked at 180 °C. They reported a high antioxidant activity in the unbaked dough as compared to baked goods. Gluten-free biscuits made with 100% of chestnut flour appeared to show a significantly higher oxidative stability, expressed as stability time, before fat oxidation compared to those obtained from a gluten-free commercial mixture (maize flour, pre-gelatinized rice flour, tapioca starch, sucrose, vegetable fibres, salt, thickening agents, guar flour, and hydroxypropylmethylcellulose). The oxidative stability of the samples may be related to the reduction of the content of unsaturated fatty acids and to the addition of antioxidant compounds from chestnut flour [56]. Barros et al. [57] found that the cooking process significantly changed the antioxidant activity of chestnuts. Also, a significant difference concerning the antioxidant activity was observed among the cultivars during cooking. Moreover, they found that in cooked chestnuts, the antioxidant activity was less dependent on the vitamin C content. Moreover, an increase in gallic acid during the cooking process was observed, confirming a high antioxidant activity of cooked chestnuts. Consequently, further studies are needed to confirm the antioxidant activity of cooked pasta with the addition of chestnut flour. Fares et al. [47] tested wheat pasta supplemented with bran and demonstrated the highest values of phenolic acids (both ferulic acid in the bound and free form and the total phenolic acids content) if the highest amount of bran was used. Their findings corresponded with the highest TEAC values for uncooked pasta. After boiling, which would enhance the extraction of bound phenolics from the food matrix, they found an increased amount of phenolic acids. The tested pasta showed an increase in the antioxidant capacity coupled with the higher amount of ferulic acid released during cooking. 

The test results showed that the antioxidant activity was positively correlated with the quantity of added chestnut fruit flour, the content of free phenolic acids, and total polyphenols. Pearson’s coefficients, exposing the relationship between the addition of chestnut flour, the content of polyphenols, and free phenolic acids and the antioxidant activity (measured after 30 min), are shown in Table 3. Very high, positive correlations were seen between the addition of chestnut fruit flour and the polyphenol content (*r* = 0.990) and between the quantity of the enriching additive and the content of free phenolic acids (*r* = 0.993). Also, Zhang et al. [48] observed significant relationships between the total phenolics and total antioxidant activities of chestnut kernels during steaming. Rocchetti et al. [4] observed a robust relationship between total phenolic content and the in vitro antioxidant capacities (Pearson’s coefficient = 0.90), assessing both the total phenolic content and the antioxidant activity of different GF flours compared with a gluten-containing counterpart (soft wheat flour).

Our research has demonstrated that our innovative gluten-free pasta with the addition of chestnut flour has the potential to become a source of polyphenolic compounds, including free phenolic acids, that are favorable for the human body. Prospectively, they can also broaden the range of gluten-free nutritionally valuable food products. Chestnut flour abounding in nutrients and naturally gluten-free is often recommended for the bakery products intended for celiac disease consumers as a quality-improving ingredient [27,58]. Besides, due to its unique composition, it enhances the color and aroma of such products [26,56]. So far, chestnut flour-enriched and gluten-free bread and biscuits have been studied most frequently [27,56,58], unlike pasta based on rice and field bean flour enriched with various quantities of chestnut fruit flour as an additive.

## 3. Materials and Methods 

### 3.1. Reagents

Standards of caffeic, gallic, ferulic, isoferulic, protocatechuic, 4-OH-benzoic, vanillic, *p*-coumaric, salicylic, and LC grade acetonitrile were purchased from Sigma–Aldrich Fine Chemicals (St. Louis, MO, USA). Gentisic and sinapic acids were from ChromaDex (Irvine, CA, USA). The LC grade water was prepared using a Millipore Direct-Q3 purification system (Bedford, MA, USA). Ethanol and methanol (reagent grade purity), used for the preparation of extracts, Folin-Ciocalteu reagents, and all analytical grade reagents were obtained from Avantor Performance Materials (Center Valley, PA, USA).

### 3.2. Preparation of Gluten-Free Pasta 

In this study, the gluten-free formula containing rice flour supplemented with field bean semolina in a ratio of 2/1 (*w/w*) was used as control pasta. Chestnuts flour (ecological farming, country of origin: Italy) was purchased from BioPlanet S.A. (Leszno, Poland) in a powdered form. The approximate composition of the flour was as follows: protein 6.4 g/100 g, fat 3.7 g/100 g, carbohydrates 71 g/100 g, including sugars 29.5 g/100 g (manufacturer’s data). To prepare chestnut flour supplemented pasta, the initial formula containing rice and field bean semolina was initially dry-mixed with different levels of chestnut flour (10, 20, 30, 40, and 50% on formula replacement basis) and processed as described follow: rice/field bean mixed semolina, salt (1%), and chestnut flour were mechanically pre-mixed for 2 min. Next, all ingredients were hydrated with the optimal amount of distilled water obtained in the preliminary study and mixed for 15 minutes at 25 °C using a KitchenAid model kPM5 (St. Joseph., MI, USA) until a homogenous dough was obtained. The divided dough was left at room temperature for 1 h, then molded and passed through reduction rolls of a pasta machine Marcato Ampia type 150 (Campodarsego, Italy) for four times per pass and in all directions to produce a uniform dough sheet [59]. The final thickness of each dough sheet was 1.5 mm, as determined with a caliper. Sheets were cut for tagliatelle shape with a width of 10 mm and 50 mm long. Finally, pasta samples were dried with an air oven with mechanical circulation at 40 °C for 8 h and stored in plastic bags at room temperature.

### 3.3. Preparation of Extracts

Two grams of pasta samples were prepared and mixed with 40 mL of ethanol. The extraction process was performed in an ultrasonic bath with a thermostat (BANDELIN electronic GmbH & Co. KG, Berlin, Germany) for 40 min at the temperature of 60 °C and the ultrasound frequency of 33 kHz and the power of 320 W. The extracts were filtered, and 40 mL of ethanol was added to the residue to repeat the extraction. The obtained extracts were mixed and evaporated. The dry residues were quantitatively transferred to volumetric flasks and refilled with methanol up to 5 mL [37,60]. 

### 3.4. Analysis of Phenolic Acids 

Phenolic acids content was determined according to the modified method described by Oniszczuk et al. [34]. Experiments were carried out using an Agilent 1200 Series HPLC system (Agilent Technologies, Santa Clara, CA, USA) connected to 3200 QTRAP Mass spectrometer (AB Sciex, Redwood City, CA, USA) equipped with electrospray ionization source (ESI). Both were controlled with Analyst 1.5 software (AB Sciex), which was also used for data interpretation. Separations were carried out on a Zorbax SB-C18 column (2.1 × 100 mm, 1.8 µm particle size; Agilent Technologies) at 20 °C. Gradient method was used with mobile phases: water with 0.1% HCOOH (A) and acetonitrile with 0.1% HCOOH (B). Injection volume was 3 µL, the flow rate was 250 µL/min, and the gradients were as follows: 0–2 min—25% B, 2–3 min—25%–35% B, 3–6 min—35% B, 6–8 min—35–55% B, 8–10 min—55% B, 10–12 min—55–75% B, 12–16 min—75% B, 16–19 min—75–25% B, 19–25 min—25% B.

ESI operated in the negative-ion mode worked in the following conditions: capillary temperature 400 °C, curtain gas at 30 psi, nebulizer gas at 50 psi, negative ionization mode source voltage −4500 V. Triplicate injections were made for each standard solution and the sample. The analytes were identified by comparing retention time and m/z values obtained by MS and MS2 with the mass spectra from the corresponding standards tested under the same conditions. The external standard method for the optimization of ESI-MS/MS parameters and quantitation was used. The control sample with all standard compounds at known concentrations was run for every sample to control instrument response and retention times. No fluctuations in the retention times or instrument response were observed. The identified phenolic acids were quantified based on their peak areas, and the comparison with calibration curves was obtained with the corresponding standards. 

### 3.5. Determination of the Total Content of Polyphenolic Compounds

The total content of polyphenolic compounds was determined with the modified Folin-Ciocalteu (FC) method [34,35]. Next, 900 μL of distilled water and 100 μL of Folin-Ciocalteu reagent were added to 100 μL of the tested extracts. The solutions were mixed and put aside. After 4 min, 1 mL of 7.7% sodium bicarbonate and 400 μL of distilled water were added. The content was mixed and placed in a water bath (40 °C) for 50 min. After that, the absorbance of the solution was measured using a UV-VIS spectrometer (Genesys 10S UV-VIS, Thermo Scientific, Waltham, MA, US; wavelength range from 190 to 1100 nm) at a wavelength of 765 nm. Then, a calibration curve of gallic acid was plotted. The 100 μL of each of the calibration solutions were collected, and 900 μL of distilled water and 100 μL of Folin-Ciocalteu reagent were added to them and processed following the same procedure as for the extracts above. A solution without gallic acid was used for the blank experiment. The total content of polyphenols in the tested extracts was expressed as GAE. 

### 3.6. Radical Scavenging Assay

The antioxidative capacity of the tested extracts was measured using a 0.1 mM methanol solution of the DPPH stable radical (2,2-diphenyl-1-picrylhydrazyl). Absorbance was measured at 517 nm wavelength, and the UV-VIS spectrophotometer (Genesys 10S UV-VIS, Thermo Scientific) was calibrated to pure methanol. In the following step, the authors measured the absorbance of the samples containing 2.5 mL of DPPH solution and 0.5 mL of the extract. The measurements were carried out every 5 min for half an hour [35]. Based on the results, the free radical scavenging ability of the tested extracts was calculated using the following formula:(1)$$\mathrm{Scavenging}\%=\frac{A_{blank}-A_{sample}}{A_{blank}}100\%$$

### 3.7. Statistical Evaluation

Statistical analysis of data was made using the MSExcel 2013 and Statistica 12.0program (StatSoft Inc., Tulsa, OK, USA). All analytical measurements were repeated in three repetitions for each sample and reference compounds. The obtained results were expressed as mean values ± SD. The significance of differences of the obtained results was determined at α < 0.05 with Duncan’s test to evaluate the homogenous groups. 

## Figures and Tables

**Figure 1 molecules-24-02623-f001:**
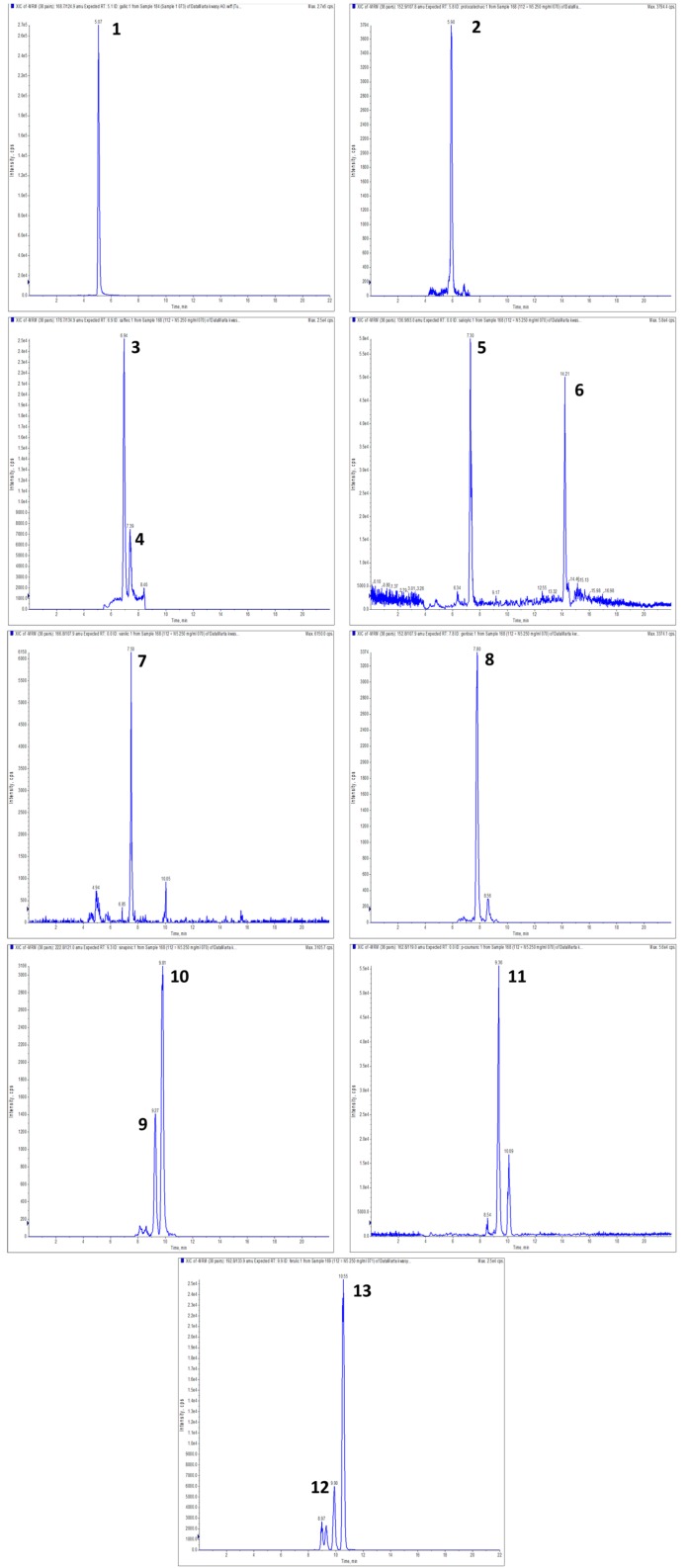
Extracted LC-MS-MRM chromatogram of phenolic acids found in gluten-free pasta, with addition of 50% chestnut flour, MRM transition are given in brackets: 1—gallic acid (m/z 168.7 → 124.9); 2—protocatechuic (m/z 152.9 → 107.8); 3—*trans*-caffeic (m/z 178.7 → 134.9); 4—*cis*-caffeic (m/z 178.7 → 134.9); 5—4-hydroxy-benzoic (m/z 136.9 → 93); 6—salicylic (m/z 136.9 → 93); 7—vanillic acid (m/z 166.8 → 107.9); 8—gentisic (m/z 152.8 → 107.9); 9—*trans*-sinapic acid (m/z 222.8 → 148.9); 10—*cis*-sinapic acid (m/z 222.8 → 148.9); 11—*p*-coumaric (m/z 162.8 → 119); 12—ferulic (m/z 192.8 → 133.9); 13—isoferulic acid (m/z m/z 192.8 → 133.9).

**Table 1 molecules-24-02623-t001:** Content of phenolic acids in gluten-free pasta samples enriched with chestnut flour (*n* = 3; mean ± SD, standard deviation).

Chestnut Flour (%)		Content of Phenolic Acids (µg/g d.w.)												
	Gallic	Proto-Catechuic	*trans*-Caffeic	*cis*-Caffeic	4-OH-benzoic	Salicylic	Vanilic	Gentisic	*trans*-Sinapic	*cis*-Sinapic	*p*-Coumaric	Ferulic	Isoferulic	Sum
0	ND	0.171^a^ ± 0.002	ND	ND	1.393^b^ ± 0.051	1.210^c^ ± 0.042	ND	0.041^a^ ± 0.0002	ND	ND	0.274^a^ ± 0.008	0.793^a^ ± 0.013	30.133^a^ ± 0.576	34.015^a^ ± 0.6922
10	BQL	0.189^ab^ ± 0.006	BQL	BQL	1.291^ab^ ± 0.043	1.020^bc^ ± 0.035	ND	0.050^ab^ ± 0.0000	BQL	BQL	0.416^ab^ ± 0.012	0.920^ab^ ± 0.033	35.040^ab^ ± 0.987	38.927^ab^ ± 1.1160
20	1.196^a^ ± 0.043	0.199^b^ ± 0.003	BQL	BQL	1.244^ab^ ± 0.009	0.944^b^ ± 0.036	BQL	0.054^ab^ ± 0.0002	BQL	0.488^a^ ± 0.008	0.636^b^ ± 0.008	1.016^ab^ ± 0.009	41.200^b^ ± 0.089	46.977^b^ ± 0.2052
30	3.516^b^ ± 0.024	0.265^bc^ ± 0.004	0.038^a^± 0.000	BQL	1.096^a^ ± 0.002	0.908^b^ ± 0.029	BQL	0.058^ab^ ± 0.0001	0.408^a^ ± 0.012	1.104^b^ ± 0.006	0.692^b^ ± 0.002	1.264^b^ ± 0.011	42.120^bc^ ± 1.220	51.468^bc^ ± 1.3101
40	5.280^c^ ± 0.006	0.335^c^ ± 0.011	0.194^b^± 0.002	BQL	1.064^a^ ± 0.038	0.716^ab^ ± 0.009	2.232^a^ ± 0.101	0.065^b^ ± 0.0001	0.520^ab^ ± 0.024	1.348^bc^ ± 0.032	0.754^bc^ ± 0.000	1.272^b^ ± 0.005	42.813^bc^ ± 0.093	56.595^c^ ± 0.3211
50	8.120^d^ ± 0.003	0.354^c^ ± 0.007	0.399^c^± 0.013	BQL	1.032^a^ ± 0.005	0.612^a^ ± 0.028	2.688^b^ ± 0.076	0.067^b^ ± 0.0002	0.736^b^ ± 0.005	1.832^c^ ± 0.067	1.428^c^ ± 0.024	1.724^c^ ± 0.052	46.000^c^ ± 0.659	65.010^d^ ± 0.9392

^a-d^—different letters in columns indicate significant differences at α=0.05; ND—not detected; BQL—below quantification level.

**Table 2 molecules-24-02623-t002:** Radical scavenging activity of gluten-free pasta samples depends on time and chestnut flour addition (*n* = 3; mean ± SD).

Radical Scavenging Towards DPPH (%)						
Time (min)	Addition of Chestnut Flour (%)					
	0	10	20	30	40	50
0	11.23^a^ ± 0.47	13.70^ab^ ± 0.01	14.63^b^ ± 0.01	16.42^bc^ ± 0.06	20.16^c^ ± 0.68	33.37^d^ ± 0.01
5	91.54^a^ ± 0.98	93.79^b^ ± 1.78	93.79^b^ ± 0.79	94.46^c^ ± 1.57	93.96^bc^ ± 2.89	94.36^c^ ± 1.69
10	92.77^a^ ± 1.23	93.95^ab^ ± 0.32	93.95^ab^ ± 1.43	94.67^b^ ± 0.57	94.56^b^ ± 1.12	94.92^b^ ± 0.56
15	92.89^a^ ± 2.67	94.06^b^ ± 0.97	94.06^b^ ± 2.11	94.66^bc^ ± 0.64	94.67^bc^ ± 1.89	95.16^c^ ± 0.00
20	93.02^a^ ± 0.12	94.11^b^ ± 0.00	94.42^bc^ ± 0.20	94.66^bc^ ± 0.08	94.71^bc^ ± 0.33	95.16^c^ ± 0.23
25	93.02^a^ ± 0.01	94.11^b^ ± 0.09	94.42^bc^ ± 0.00	94.66^bc^ ± 0.07	94.71^bc^ ± 0.04	95.16^c^ ± 0.13
30	93.02^a^ ± 0.03	94.11^b^ ± 0.05	94.42^bc^ ± 0.00	94.66^bc^ ± 0.01	94.71^bc^ ± 0.31	95.16^c^ ± 0.00

^a–d^—different letters in rows indicate significant differences at α = 0.05.

**Table 3 molecules-24-02623-t003:** Pearson’s correlation coefficients (at α < 0.05).

	Total Polyphenols	Free Phenolic Acid	DPPH Radical Scavenging Capacity
Chestnut content	0.990	0.993	0.982
Total polyphenols		0.977	0.963
Free phenolic acids			0.976

DPPH–2,2-diphenyl-1-picrylhydrazyl.

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
