# Peer review of "Content of Phenolic Compounds and Antioxidant Activity of New Gluten-Free Pasta with the Addition of Chestnut Flour"

_molecules, 2019, doi:10.3390/molecules24142623_

Round 1
Reviewer 1 Report
Manuscript is very interesting, pointing out possible application of new gluten‐free pasta with addition of
chestnut flour.
The manuscript fit the journal's topic, the topic is actual and the use Recent Advances in Food Chemistry and Microbiology in Relation to Health in this way is attractive and of interest.
The manuscript is quite well organised and rationally drawn.
Main weakness are in my opinion related to the scarce literature references expecially in the assingment of several compounds by mass spectrometry.
I suggest a revision of the part related to the assignments of compounds.
Author Response
Answers to Reviewers’ comments
The authors would like to thank all Reviewers for your valuable comments. The paper has been corrected according to remarks and suggestions of Reviewers.
Reviewer 1
Manuscript is very interesting, pointing out possible application of new gluten‐free pasta with addition of chestnut flour. The manuscript fit the journal's topic, the topic is actual and the use Recent Advances in Food Chemistry and Microbiology in Relation to Health in this way is attractive and of interest. The manuscript is quite well organised and rationally drawn. Main weakness are in my opinion related to the scarce literature references expecially in the assingment of several compounds by mass spectrometry.
I suggest a revision of the part related to the assignments of compounds.
Additional discussion has been added as requested, literature review has been extended as well as discussion of the results.
Reviewer 2 Report
Comments
This manuscript described the “Content of phenolic compounds and antioxidant activity of new gluten-free pasta with addition of chestnut flour”, it is an interesting study. Recently, the authors have published a paper titled “The Impact of Processing Parameters on the Content of Phenolic Compounds in New Gluten-Free Precooked Buckwheat Pasta” in Molecules using almost the same methods employed in the current manuscript. However, in my opinion, the novelty of this manuscript is low and cannot be published in Molecules. In addition, some low-level mistakes also exist.
Some questions and suggestions are listed as follows:
1) HPLC-ESI-MS/MS means “high performance liquid chromatography ionization electrospray tandem mass spectrometry” rather than “high performance liquid chromatography coupled with mass spectrometry”.
2) Figure 1 must be revised carefully and it is not suitable for publication now. In the figure caption, “m/z 168.7 > 124.9” should be revised as “m/z 168.7→124.9”.
3) The content of free phenolic acids was listed in Table 1, but I cannot find the content of total polyphenols in any tables or figures.
4) Page 6, line143-148. The authors said that “Pearson’s coefficients, exposing the relationship between the addition of chestnut flour, the content of polyphenols, flavonoids, and free phenolic acids and the antioxidant activity (measured after 30 minutes), are shown in Table 3. Very high, positive correlations were seen between the addition of chestnut fruit flour and the polyphenol content (r = 0.993) and between the quantity of the enriching additive and the content of free phenolic acids (r = 0.990).” However, flavonoids cannot be found in Table 3. More importantly, in Table 3, the correlation between chestnut content and total polyphenols is 0.990 rather than 0.993, and the correlation between chestnut content and the content of free phenolic acids is 0.993 rather than 0.990.
5) Page 7, line 202. For a Zorbax SB‐C18 column used in this study, “1.8-mm particle size” must be a serious error. Whether 1.8 μm?
6) Page 8, line 204. The author said that “The flow rate of mobile phase was 250 mL/min”, this must be also a serious error, 250 mL/min is impossible HPLC-ESI-MS/MS!!!
7) Page 8, line 205. The authors said that “the gradients were as follows: 0-2 min ‐ 25% B, 3‐6 min ‐ 35% B, 8‐10 min ‐ 55% B, 12‐16 min ‐ 75% B, 19‐25 min ‐ 25% B.” What about “2-3 min, 6-8 min, 10-12 min”? The mobile gradients must be continuous.
8) What is the internal standard in the HPLC-ESI-MS/MS analysis? I cannot find it or them through the whole manuscript.
9) Page 8, line 219-221. The authors said that “After that, the absorbance of the solution was measured using a UV spectrometer (Genesys 10S VIS, Thermo Scientific, Waltham, MA, US) at a wavelength of 765 nm.” For UV spectrometer, the range of wavelength is from 10-400 nm, I want to know why a wavelength of 765 nm can be obtained.
10) In summary, I wonder whether this quantitative analysis is performed by the authors?
Author Response
The authors would like to thank all Reviewers for your valuable comments. The paper has been corrected according to remarks and suggestions of Reviewers.
Reviewer 2
This manuscript described the “Content of phenolic compounds and antioxidant activity of new gluten-free pasta with addition of chestnut flour”, it is an interesting study. Recently, the authors have published a paper titled “The Impact of Processing Parameters on the Content of Phenolic Compounds in New Gluten-Free Precooked Buckwheat Pasta” in Molecules using almost the same methods employed in the current manuscript. However, in my opinion, the novelty of this manuscript is low and cannot be published in Molecules. In addition, some low-level mistakes also exist.
Authors used similar methods employed in the manuscript concerning precooked buckwheat pasta. However, the food was made from completely different raw materials and using different production methods. Buckwheat pasta was produced using the extrusion-cooking method with the high temperature treatment and pasta was ready for consumption after hot water hydration without cooking, but the gluten-free pasta with addition of chestnut flour presented in this study was made manually as laminated sheeted pasta.
Some questions and suggestions are listed as follows:
1) HPLC-ESI-MS/MS means “high performance liquid chromatography ionization electrospray tandem mass spectrometry” rather than “high performance liquid chromatography coupled with mass spectrometry”.
The proper change has been now made in the text.
2) Figure 1 must be revised carefully and it is not suitable for publication now. In the figure caption, “m/z 168.7 > 124.9” should be revised as “m/z 168.7→124.9”.
Figure has been improved and figure caption has been now corrected.
3) The content of free phenolic acids was listed in Table 1, but I cannot find the content of total polyphenols in any tables or figures.
The content of total polyphenols has been mentioned in the text; lines 151-152: “Total polyphenols showed the values of 2.123, 2.621, 2.955, 3.886, 4.816 and 5.721 mg GAE/g of dry matter for the control sample and samples enriched with the addition of 10, 20, 30, 40 and 50% of chestnut flour, respectively.”
4) Page 6, line143-148. The authors said that “Pearson’s coefficients, exposing the relationship between the addition of chestnut flour, the content of polyphenols, flavonoids, and free phenolic acids and the antioxidant activity (measured after 30 minutes), are shown in Table 3. Very high, positive correlations were seen between the addition of chestnut fruit flour and the polyphenol content (r = 0.993) and between the quantity of the enriching additive and the content of free phenolic acids (r = 0.990).” However, flavonoids cannot be found in Table 3. More importantly, in Table 3, the correlation between chestnut content and total polyphenols is 0.990 rather than 0.993, and the correlation between chestnut content and the content of free phenolic acids is 0.993 rather than 0.990.
We apologize for our error. The authors accidentally attached an outdated version of the table. In addition, we inattentively described the results. Total flavonoids content was not determined in this study. It is now corrected in the manuscript.
5) Page 7, line 202. For a Zorbax SB‐C18 column used in this study, “1.8-mm particle size” must be a serious error. Whether 1.8 μm?
6) Page 8, line 204. The author said that “The flow rate of mobile phase was 250 mL/min”, this must be also a serious error, 250 mL/min is impossible HPLC-ESI-MS/MS!!!
5) and 6) We apologize for our neglect. The authors did not check carefully the manuscript after a native speaker correction, that is why these errors appeared in the text (1.8-mm instead of 1.8 μm and 250 mL/min instead of 250 µL/min). It has been corrected for 1.8 µm particle size and the flow rate was 250 µL/min.
7) Page 8, line 205. The authors said that “the gradients were as follows: 0-2 min ‐ 25% B, 3‐6 min ‐ 35% B, 8‐10 min ‐ 55% B, 12‐16 min ‐ 75% B, 19‐25 min ‐ 25% B.” What about “2-3 min, 6-8 min, 10-12 min”? The mobile gradients must be continuous.
Thank you for your suggestion. The authors rewrote the sentence and corrected the method description. Now is: “Injection volume was 3 µL, the flow rate was 250 µL/min and the gradients were as follows: 0-2 min – 25%B, 2-3 min – 25%-35%B, 3-6 min – 35%B, 6-8 min – 35-55%B, 8-10 min – 55%B, 10-12 min 55-75%B, 12-16 min - 75%B, 16-19 min – 75-25%B, 19-25 min - 25%B.”
8) What is the internal standard in the HPLC-ESI-MS/MS analysis? I cannot find it or them through the whole manuscript.
We have used external standard method for optimization of ESI-MS/MS parameters and quantitation. Control sample with all standard compounds at known concentrations was run every 12th sample to control instrument response and retention times. No fluctuations of retention times or in instrument response were observed. The proper information has been added to the text.
9) Page 8, line 219-221. The authors said that “After that, the absorbance of the solution was measured using a UV spectrometer (Genesys 10S VIS, Thermo Scientific, Waltham, MA, US) at a wavelength of 765 nm.” For UV spectrometer, the range of wavelength is from 10-400 nm, I want to know why a wavelength of 765 nm can be obtained.
We apologize for an error. The absorbance of the solution was measured using a UV-VIS spectrometer (Genesys 10S UV-VIS, Thermo Scientific, Waltham, MA, US; wavelength range from 190 to 1100 nm). The effort has been corrected in the text. The same spectrometer was used in all analysis (3.5. Determination of the total content of polyphenolic compounds and 3.6. Radical scavenging assay).
10) In summary, I wonder whether this quantitative analysis is performed by the authors?
The authors assure that they carried out all the analysis described in the manuscript.
Reviewer 3 Report
Summary
The paper aims to describe the content of phenolic compounds and the antioxidant activity of a gluten free pasta added with different percentages of chestnut flour. The paper reports no clear information about the background to understand the novelty of the research compared to the existing ones.
Broad comments
The results of the study describe the total polyphenol content and the antioxidant properties of a gluten free pasta with the addition of different doses of chestnut flour. Moreover, the phenolic acids present are described qualitatively and quantitatively. These assessments are important considering the beneficial health effects of polyphenols and the possibility of developing a gluten-free product with good nutritional characteristics. However, evaluations have been made only on uncooked pasta and not on cooked pasta, so it is not known if cooking can cause a loss of polyphenols. It is not described how the presence of chestnut flour can affect other characteristics of pasta, eg mechanical characteristics, taste, appearance, nor it is reported if other studies are planned or necessary to evaluate these aspects. There are no comparisons of the results obtained with literature data. The introduction is extremely lacking in the description of the background and in the reasons behind the study.
Specific comments
Abstract
Line 19: the description of the nutritional properties of chestnuts is to be improved, proteins are macronutrients and pectins are fibers, they should not be treated separately
Line 30: English… “have a potential to be” must be replaced with “has a potential to be”
Line 32: “essential for human health” is generally attributable to essential fatty acids or essential amino acids. Are there any literature data available where phenolic acids are defined as essential?
Introduction
The introduction doesn’t accurately describe the current knowledge related to the research question neither the background: no informations are reported about gluten free products, phenolic compounds in gluten free products and the description of nutritional properties of chestnut is poor. It contains unnecessary information as the ones related to Algerian cuisine. The reasons for performing the study are not clear: it is not clear why chestnut flour has been tested in a gluten-free pasta: if the goal was just to create a product rich in polyphenols, health promoting and tasty (as mentioned after in results and discussion), why not try it in pasta with wheat flour?
Lines 38-48: the connection between the description made of Algerian cuisine and the subject of the article is not understandable
Lines 60-62: the description of the nutritional properties of the chestnut must be improved, citing also the fact that there are different cultivars with different characteristics. Starch content is also important if it’s considered the use for a gluten-free product. Protein quality has to be described more correctly; one of the aforementioned references (ie n. 8) reports as follows about essential amino acids: “In general, chestnuts are a good source of these compounds, however, amino acids profiles are not well balanced, with certain essential amino acids occurring in limiting concentration when compared to FAO (1990) recommended levels”.
Matherials and Methods
Which cultivar of chestnut was used?
Results and discussions
Findings of the study are not compared at all with the findings of other studies on similar products: paragraphs 2.1 and 2.2 should be completed with comparison with literature data.
Other characteristics of the pasta gluten free with chestnut flour considered relevant for its use by consumers and which have to be studied have to be mentioned: i.e. mechanical properties, taste, behavior after coking…
Limitations of the study have to be mentioned. One limitation is the fact that no consideration has been given to the analysis of cooked pasta: some nutrients can be lost thorough cooking.
Line 102: is chestnut flour a functional additive?
Lines 112-113: when it is reported “the obtained pasta are a great source of free phenolic acids”, a comparison should be made with polyphenol content in other types of pasta
Lines 131 and 137 : is chestnut flour a functional additive?
Line 153: “essential for human body” is generally attributable to essential fatty acids or essential amino acids. Are there any literature data available where phenolic acids are defined as essential?
Lines 154-152: description of gluten-free products market and of gluten-free products with chestnut flour should be presented in the introduction instead of the discussion and broadened.
Tables 1 e 2: the letters that indicate significant differences should be explained.
Author Response
The authors would like to thank all Reviewers for your valuable comments. The paper has been corrected according to remarks and suggestions of Reviewers.
Reviewer 3
Summary
The paper aims to describe the content of phenolic compounds and the antioxidant activity of a gluten free pasta added with different percentages of chestnut flour. The paper reports no clear information about the background to understand the novelty of the research compared to the existing ones.
The proper description has been added in Introduction section to highlight the novelty of presented paper.
Broad comments
The results of the study describe the total polyphenol content and the antioxidant properties of a gluten free pasta with the addition of different doses of chestnut flour. Moreover, the phenolic acids present are described qualitatively and quantitatively. These assessments are important considering the beneficial health effects of polyphenols and the possibility of developing a gluten-free product with good nutritional characteristics. However, evaluations have been made only on uncooked pasta and not on cooked pasta, so it is not known if cooking can cause a loss of polyphenols. It is not described how the presence of chestnut flour can affect other characteristics of pasta, eg mechanical characteristics, taste, appearance, nor it is reported if other studies are planned or necessary to evaluate these aspects. There are no comparisons of the results obtained with literature data. The introduction is extremely lacking in the description of the background and in the reasons behind the study.
The paper describes first time ever application of chestnut flour to improve nutritional value of gluten-free laminated pasta based on rice-field bean mixture so it is lack of reports for discussion of the results. In Scopus database we have found only 1 work of Kosović and coworkers [2016] with chestnut flour application in pasta but durum wheat-based pasta was prepared, and thus the product is not suitable for celiac consumers. Introduction section has been extended to highlight the novelty of presented study. We have added some discussion about composition and functional components of other products with chestnuts added as well as other types of pasta. Additionally, some discussion has been added according to treatment effect on polyphenols and antioxidant activity of various products. The aim of the paper was identification of phenolic compounds and antioxidant activity of developed dry pasta, other results (as cooking quality, texture, sensory and cooking pasta composition) will be presented as a separate study soon.
Specific comments
Abstract
Line 19: the description of the nutritional properties of chestnuts is to be improved, proteins are macronutrients and pectins are fibers, they should not be treated separately
Changed for: Chestnut fruit abounds in proteins, unsaturated fatty acids, fiber, polyphenolic compounds as well as vitamins and micronutrients that are behind the health-promoting properties of this plant
Line 30: English… “have a potential to be” must be replaced with “has a potential to be”
Changed for: has a potential to be
Line 32: “essential for human health” is generally attributable to essential fatty acids or essential amino acids. Are there any literature data available where phenolic acids are defined as essential?
It has been changed for valuable.
Introduction
The introduction doesn’t accurately describe the current knowledge related to the research question neither the background: no informations are reported about gluten free products, phenolic compounds in gluten free products and the description of nutritional properties of chestnut is poor. It contains unnecessary information as the ones related to Algerian cuisine. The reasons for performing the study are not clear: it is not clear why chestnut flour has been tested in a gluten-free pasta: if the goal was just to create a product rich in polyphenols, health promoting and tasty (as mentioned after in results and discussion), why not try it in pasta with wheat flour?
Additional information about gluten-free products as well as phenolic compounds in gluten-free products has been added in Introduction as well as in Results and discussion section.
Lines 38-48: the connection between the description made of Algerian cuisine and the subject of the article is not understandable
Introduction section has been changed to avoid misunderstandings because not only Algerian consumers are looking for new gluten-free products with increased nutritional value.
Lines 60-62: the description of the nutritional properties of the chestnut must be improved, citing also the fact that there are different cultivars with different characteristics. Starch content is also important if it’s considered the use for a gluten-free product. Protein quality has to be described more correctly; one of the aforementioned references (ie n. 8) reports as follows about essential amino acids: “In general, chestnuts are a good source of these compounds, however, amino acids profiles are not well balanced, with certain essential amino acids occurring in limiting concentration when compared to FAO (1990) recommended levels”.
Proper information has been added.
Matherials and Methods
Which cultivar of chestnut was used?
Proper information has been added
Results and discussions
Findings of the study are not compared at all with the findings of other studies on similar products: paragraphs 2.1 and 2.2 should be completed with comparison with literature data.
Proper discussion has been added.
Other characteristics of the pasta gluten free with chestnut flour considered relevant for its use by consumers and which have to be studied have to be mentioned: i.e. mechanical properties, taste, behavior after coking…
Other characteristics were not presented in this paper to fit the scope of the Molecules journal, additional paper is now prepared according to cooking quality of chestnut supplemented pasta.
Limitations of the study have to be mentioned. One limitation is the fact that no consideration has been given to the analysis of cooked pasta: some nutrients can be lost thorough cooking.
Proper comments have been added in Results and discussion section.
Line 102: is chestnut flour a functional additive?
Authors didn’t used that description, but: “The tests showed that the obtained pastas are a great source of free phenolic acids as natural antioxidants. It has been widely demonstrated that the consumption of foods with a high content of phenolic acids promises anti-cancer, anti-bacterial and anti-inflammatory effects.”
Lines 112-113: when it is reported “the obtained pasta are a great source of free phenolic acids”, a comparison should be made with polyphenol content in other types of pasta
Proper discussion has been added as requested.
Lines 131 and 137 : is chestnut flour a functional additive?
We rewrite these sentences: “Our research has demonstrated that our innovative gluten-free pasta with the addition of chestnut flour have a potential to become a source of polyphenolic compounds, including free phenolic acids that are favorable for the human body. Prospectively, they can also broaden the range of gluten-free nutritionally valuable food products.”
Line 153: “essential for human body” is generally attributable to essential fatty acids or essential amino acids. Are there any literature data available where phenolic acids are defined as essential?
It has been changed for: favorable
Lines 154-152: description of gluten-free products market and of gluten-free products with chestnut flour should be presented in the introduction instead of the discussion and broadened.
Proper changed have been made.
Tables 1 e 2: the letters that indicate significant differences should be explained.
Explanations are presented as footnotes under table 1: a-d – different letters in columns indicate significant differences at α=0.05; ND – not detected; BQL – below quantification level
and 2: a-d – different letters in rows indicate significant differences at α=0.05
Round 2
Reviewer 2 Report
HPLC-ESI-MS/MS means high performance liquid chromatography electrospray ionization tandem mass spectrometry. In addition, the figures should be further improved. All the figures and tables should be embeded in corresponding section rather than a single section as "Figures and Tables"
Author Response
Reviewer 2
HPLC-ESI-MS/MS means high performance liquid chromatography electrospray ionization tandem mass spectrometry.
It has been corrected.
In addition, the figures should be further improved. All the figures and tables should be embeded in corresponding section rather than a single section as "Figures and Tables"
Tables and figure have been embedded in the text, as requested.
Reviewer 3 Report
The quality of the paper has improved, however some aspects of the introduction and the results need some revisions.
Abstract
Line 19: It is important to mention the main nutritional characteristic of chestnut that is the presence of carbohydrates, mostly starch. You should add the word carbohydrates or starch.
Introduction
Te first part of the introduction is still not relevant to the subject of the study. It is not clear why it is mentioned the tradition of preparing pasta, unless the purpose of the research is to provide a recipe for making pasta at an artisan-homemade level. I would eliminate lines 38 to 46.
I think that introduction should first focus on celiac disease and gluten-free diet and products (lines 76-107). Subsequently functional foods can be considered (lines 46-55), synthesizing very much and modifying the above comparing it to gluten-free products. After chestnut fruit characteristics can be introduced (lines 56-75).
Line 62: explain WW2
Line 67: you should add cultivar: may vary depend on cultivar, origin, country etc.
Line 73: minerals are micronutrients not macronutrients, please correct
Matherials and Methods
Can you trace the cultivar of the chestnut flour used?
Results
Lines 144-146: specify the type of pasta
Lines 146-161: here is described the effect of thermal treatments on quality and quantity of phenolic acids on pasta. I think that this topic should be summarized and dealt with later, introduced by some considerations on the fact that it is important to consider the effects of the treatments following the preparation of the pasta (eg cooking) on the quali-quantitative content of phenolic acids.
Line 166: instead of reporting phenolics content in sweet chestnut you should report phenolics content in pasta, specifying the type of pasta
Line 173-174: that phrase is still not justified because there are no comparison data on the content of phenolic acids in other types of pasta, indeed in line 45 it says that the content found is similar to values reported previously
Lines 178-182: I think that this topic should be dealt with later, introduced by some considerations on the fact that it is important to consider the effects of chestnut flour on mechanical properties of pasta
Line 197: functional additive?
Lines 2016-207: “Gluten-free biscuits produced with chestnut flour appeared higher oxidative stability values” not so clear, higher respect to which product? Oxidative stability?
Line 228: specify the food product

Author Response
Reviewer 3
The quality of the paper has improved, however some aspects of the introduction and the results need some revisions.
Abstract
Line 19: It is important to mention the main nutritional characteristic of chestnut that is the presence of carbohydrates, mostly starch. You should add the word carbohydrates or starch.
“carbohydrates” word has been added.
Introduction
The first part of the introduction is still not relevant to the subject of the study. It is not clear why it is mentioned the tradition of preparing pasta, unless the purpose of the research is to provide a recipe for making pasta at an artisan-homemade level. I would eliminate lines 38 to 46.
The mentioned sentences have been removed.
I think that introduction should first focus on celiac disease and gluten-free diet and products (lines 76-107). Subsequently functional foods can be considered (lines 46-55), synthesizing very much and modifying the above comparing it to gluten-free products. After chestnut fruit characteristics can be introduced (lines 56-75).
Introduction has been rewritten according to Reviewer’s suggestions.
Line 62: explain WW2
It was short of World War II, we’ve changed for Second World War
Line 67: you should add cultivar: may vary depend on cultivar, origin, country etc.
It has been added
Line 73: minerals are micronutrients not macronutrients, please correct
It has been corrected
Matherials and Methods
Can you trace the cultivar of the chestnut flour used?
The chestnut flour was bought via internet shop, distributor is giving only information about “ecological farming” and proximate chemical composition, so we added these details in MM section, it should be helpful to compare the results with other raw materials.
Results
Lines 144-146: specify the type of pasta
Explanations have been added with proper references.
Lines 146-161: here is described the effect of thermal treatments on quality and quantity of phenolic acids on pasta. I think that this topic should be summarized and dealt with later, introduced by some considerations on the fact that it is important to consider the effects of the treatments following the preparation of the pasta (eg cooking) on the quali-quantitative content of phenolic acids.
The proper changes have been made
Line 166: instead of reporting phenolics content in sweet chestnut you should report phenolics content in pasta, specifying the type of pasta
There were some additional citations added
Line 173-174: that phrase is still not justified because there are no comparison data on the content of phenolic acids in other types of pasta, indeed in line 45 it says that the content found is similar to values reported previously
Proper previously cited results have been added
Lines 178-182: I think that this topic should be dealt with later, introduced by some considerations on the fact that it is important to consider the effects of chestnut flour on mechanical properties of pasta
It has been changed as requested
Line 197: functional additive?
It has been changed for “additive”
Lines 2016-207: “Gluten-free biscuits produced with chestnut flour appeared higher oxidative stability values” not so clear, higher respect to which product? Oxidative stability?
It has been corrected for: “Gluten-free biscuits made with 100% of chestnut flour appeared to show significantly higher oxidative stability, expressed as “stability time” before fat oxidation, if compared to those obtained from gluten-free commercial mixture (maize flour, pre-gelatinized rice flour, tapioca starch, sucrose, vegetable fibres, salt, thickening agents (guar flour and hydroxypropylmethylcellulose). Oxidative stability of the samples may be related to the reduction of the unsaturated fatty acids' content and to the addition of antioxidant compounds from chestnut flour [54].
Line 228: specify the food product
It has been added: “of chestnuts kernels during steaming.”